# Functions of Osteocalcin in Bone, Pancreas, Testis, and Muscle

**DOI:** 10.3390/ijms21207513

**Published:** 2020-10-12

**Authors:** Toshihisa Komori

**Affiliations:** Basic and Translational Research Center for Hard Tissue Disease, Nagasaki University Graduate School of Biomedical Sciences, Nagasaki 852-8588, Japan; komorit@nagasaki-u.ac.jp; Tel.: +81-95-819-7637; Fax: +81-95-819-7638

**Keywords:** osteocalcin, apatite crystal, collagen, bone formation, bone strength, glucose metabolism, testosterone, muscle

## Abstract

Osteocalcin (Ocn), which is specifically produced by osteoblasts, and is the most abundant non-collagenous protein in bone, was demonstrated to inhibit bone formation and function as a hormone, which regulates glucose metabolism in the pancreas, testosterone synthesis in the testis, and muscle mass, based on the phenotype of Ocn^−/−^ mice by Karsenty’s group. Recently, Ocn^−/−^ mice were newly generated by two groups independently. Bone strength is determined by bone quantity and quality. The new Ocn^−/−^ mice revealed that Ocn is not involved in the regulation of bone formation and bone quantity, but that Ocn regulates bone quality by aligning biological apatite (BAp) parallel to the collagen fibrils. Moreover, glucose metabolism, testosterone synthesis and spermatogenesis, and muscle mass were normal in the new Ocn^−/−^ mice. Thus, the function of Ocn is the adjustment of growth orientation of BAp parallel to the collagen fibrils, which is important for bone strength to the loading direction of the long bone. However, Ocn does not play a role as a hormone in the pancreas, testis, and muscle. Clinically, serum Ocn is a marker for bone formation, and exercise increases bone formation and improves glucose metabolism, making a connection between Ocn and glucose metabolism.

## 1. Introduction

Osteocalcin (Ocn) is specifically expressed in osteoblasts and is the most abundant non-collagenous protein in bone. Ocn acquires a high affinity to Ca^2+^ by carboxylation of three glutamic acids [1,2,3,4]. Before the generation of Ocn-deficient (Ocn^−/−^) mice, carboxylated Ocn was implicated in bone mineralization [5,6,7,8]. Carboxylated Ocn was also reported to inhibit hydroxyapatite growth in mineralization [2,9,10,11]. Furthermore, it was found to function as a chemoattractant of osteoclast precursors [12,13,14]. However, Ocn^−/−^ mice exhibited different phenotypes, which were a marked increase in trabecular and cortical bone, and increased bone formation and resorption [15]. Therefore, Ocn was considered to be a bone matrix protein that inhibits bone formation and resorption. 

Moreover, Ocn was reported to function as a hormone that regulates glucose metabolism, testosterone synthesis, muscle mass, brain development and functions, and parasympathetic tone, establishing a new concept that bone and many organs, including the pancreas, testis, muscle, brain, and autonomic nervous system, are linked by the bone-derived hormone Ocn [16,17,18,19,20]. These findings were obtained using one Ocn^−/−^ mouse line generated by Karsenty’s group. Recently, our and Williams’ groups independently generated Ocn^−/−^ mice and reported different phenotypes from those of Ocn^−/−^ mice generated by Karsenty’s group. In this review, the differences in phenotypes between the previous Ocn^−/−^ mouse line and two newly established Ocn^−/−^ mouse lines are clarified, and the actual function of Ocn is discussed.

## 2. Ocn and *Runx2*

Mice have a gene cluster of osteocalcin that consists of *Bglap*, *Bglap2*, and *Bglap3* within a 23-kb span of genomic DNA, whereas one osteocalcin gene (*BGLAP*) has been identified in humans and rats [21,22,23]. *Bglap* and *Bglap2* (referred to as Ocn) are specifically expressed in osteoblasts in bone, whereas *Bglap3* is expressed in non-osteoid tissues, including the kidneys, lungs, and male gonadal tissues [23,24]. The expression of Ocn is regulated by runt related transcription factor 2 (*Runx2*), which is an essential transcription factor for osteoblast differentiation [25]. Indeed, *Runx2*^−/−^ mice express no Ocn; its expression was reduced by antisense oligonucleotides of *Runx2* in rat primary osteoblasts and ROS17/2.8 osteoblastic cells, and overexpression of *Runx2* induced Ocn expression in C3H10T1/2 multipotent mesenchymal cells [26,27,28,29]. Although *Runx2* was reported to induce Ocn in mouse skin fibroblasts strongly [28], this was unreproducible (unpublished observation), and no other reports described the induction of Ocn expression by *Runx2* in skin fibroblasts. *Runx2* is, therefore, not sufficient to induce osteoblastic differentiation and Ocn expression in skin fibroblasts. 

In addition, transgenic mice expressing a dominant-negative (dn) *Runx2*, which contains only the *Runx2* DNA-binding domain but lacks the transcriptional activation domain, at a similar expression level to endogenous *Runx2* under the control of the Ocn promoter, exhibited a markedly reduced bone volume and expressed virtually no Ocn [30]. These findings were also controversial. Transgenic mice expressing the dominant-negative *Runx2* at a much higher expression level than that of endogenous *Runx2* under the control of a 2.3-kb *Col1a1* promoter exhibited an age-dependent increase in trabecular bone due to reduced bone resorption, and Ocn expression was mildly reduced at 4 weeks of age but normal at 10 weeks of age [31]. Although both the Ocn promoter and 2.3-kb *Col1a1* promoter drive transgene expression in osteoblasts, the Ocn promoter is active in more mature osteoblasts [32], and the 2.3-kb *Col1a1*-dn*Runx2* transgenic mice expressed the transgene at a much higher level than Ocn-dn*Runx2* transgenic mice. Thus, 2.3-kb *Col1a1*-dn*Runx2* transgenic mice were expected to exhibit osteopenia and a further reduction in Ocn expression, but this was not the case. The marked reduction in bone mass in Ocn-dn*Runx2* transgenic mice was explained by the extremely strong affinity of dn*Runx2* to the *Runx2* binding sequence and the auto-positive regulation of its own promoter by *Runx2* [30]. However, we found that dn*Runx2* binds the *Runx2* recognition sequence with a similar affinity to wild-type *Runx2* and dn*Runx2* dose-dependently inhibits *Runx2* transcriptional activity [31] (unpublished observation). Furthermore, *Runx2* was demonstrated to regulate its own promoter negatively [33,34]. Thus, the marked reduction in bone mass and the absence of Ocn expression in Ocn-dn*Runx2* transgenic mice cannot be explained by the affinity of dn*Runx2* and the auto-positive regulation of its own promoter. Although there are two reports of *Runx2* conditional knockout mice using the 2.3-kb *Col1a1* promoter Cre mice, both reports did not examine the expression of bone matrix protein genes, including Ocn [35,36]. Thus, the role of *Runx2* in Ocn expression in osteoblasts in vivo remains to be clarified. 

## 3. Ocn and Bone Formation and Resorption

Ocn^−/−^ mice generated by Karsenty’s group exhibited an increase in trabecular and cortical bone [15]. These mice were analyzed in a 129Sv:C57BL/6J mixed genetic background [15]. Although quantitative data were not presented for the trabecular bone, the cortical thickness reached 150% of that in control mice without abnormal histological appearance. Bone histomorphometric analysis revealed increased bone formation in both trabecular and cortical bone in Ocn^−/−^ mice. As the osteoblast surface was not increased in Ocn^−/−^ mice, osteoblast function was considered to be increased in Ocn^−/−^ mice. The osteoclast number and bone marrow area were higher in Ocn^−/−^ mice than in wild-type mice. In addition, ovariectomy increased the bone marrow area and reduced bone strength in Ocn^−/−^ mice to a greater degree than in wild-type mice, suggesting that osteoclasts with normal function increased in Ocn^−/−^ mice [15]. Thus, Ocn^−/−^ mice generated by Karsenty’s group indicated that Ocn inhibits bone formation by inhibiting osteoblast function and inhibits bone resorption by suppressing osteoclastogenesis, demonstrating it to be a negative regulator of bone formation and resorption [15] (Table 1).

The same Ocn^−/−^ mouse line in a C57BL/6J genetic background exhibited different phenotypes. The cortical thickness and area were similar to those in wild-type mice in both males and females [37,38]. Trabecular bone and the serum level of the bone formation marker P1NP increased, but the level of the bone resorption marker C-terminal telopeptide crosslink of type I collagen (CTX1) was similar between Ocn^−/−^ mice and wild-type mice. The osteoblast number, but not osteoclast number, in Ocn^−/−^ mice increased, and the bone marrow area was similar to that in wild-type mice [38]. As the phenotypes of Ocn^−/−^ mice were different between genetic backgrounds, these two Ocn^−/−^ mouse lines with different genetic backgrounds need to be directly compared (Table 1). 

Ocn^−/−^ mice were recently generated by Williams’ group using the CRISPRA/Cas9 system. These mice were analyzed in a C57BL/6J;C3H mixed genetic background. All parameters for trabecular bone and cortical bone on micro-computed tomography (CT) analyses were similar to those in wild-type mice in both males and females [39]. Our Ocn^−/−^ mice were generated using embryonic stem (ES) cells and analyzed in a C57BL/6N genetic background. Micro-CT analyses at 14 weeks and 6 and 9 months of age revealed that bone volumes of trabecular and cortical bone were similar to those in wild-type mice in both males and females [40]. Furthermore, based on bone histomorphometric analysis, the parameters for osteoblasts, osteoclasts, and bone formation were similar between our Ocn^−/−^ mice and wild-type mice in both trabecular and cortical bone [40]. Moreover, serum markers for bone formation (N-terminal propeptide of type I procollagen: P1NP) and resorption (tartrate-resistant acid phosphatase 5b: TRAP5b and CTX1), and the expression of osteoblast and osteoclast marker genes were similar between our Ocn^−/−^ mice and wild-type mice [40]. In Ocn^−/−^ rats, the cortical bone parameters were similar to those in wild-type rats, but trabecular bone was slightly increased on micro-CT analysis [41]. Bone phenotypes were different between the Ocn^−/−^ mouse line generated by Karsenty’s group and the two Ocn^−/−^ mouse lines by Williams’ and our groups. Ocn protein was completely absent in the three Ocn^−/−^ mouse lines and Ocn^−/−^ rats [15,39,40,41]. Whether the differences in bone phenotypes are due to the genetic background of Ocn^−/−^ mouse lines needs to be investigated. The direct comparison of these Ocn^−/−^ mouse lines will resolve the controversy, especially why the Ocn^−/−^ mouse line in the 129Sv; C57BL/6J mixed genetic background exhibits a marked increase in cortical bone [15] (Table 1). The analysis of Ocn^−/−^ mouse line in a pure 129Sv genetic background may also help resolve the controversy. 

## 4. Osteocalcin and Bone Quality

The bone mineral densities (BMDs) were inconsistent among Ocn^−/−^ mouse lines. In the micro-CT analysis of the Ocn^−/−^ mouse line in the C57BL/6J genetic background by Karsenty’s group, Bailey et al. found that the cortical BMD was similar to that in wild-type, whereas Berezovska et al. noted reduced cortical BMDs [37,38]. The Ocn^−/−^ mouse line in a C57BL/6J;C3H mixed genetic background by Williams’ group and our Ocn^−/−^ mouse line in a C57BL/6N genetic background had similar cortical and trabecular BMDs to wild-type mice [39,40] (Table 1). 

The functions of Ocn for bone quality were examined in the three Ocn^−/−^ mouse lines by Fourier transform infrared microspectroscopy (FTIRM) or Raman microscopy. In the analysis of cortical bone in the Ocn^−/−^ mouse line in a mixed genetic background by Karsenty’s group using FTIRM, the crystals were smaller/less perfect than those in wild-type mice, and the mineral:matrix ratio and carbonate:phosphate ratio were similar between the Ocn^−/−^ mouse line and wild-type mice [42]. However, in the analysis of cortical bone in the same Ocn^−/−^ mouse line in a C57BL/6J genetic background using Raman microscopy, the crystallinity was increased, carbonate:phosphate ratio was reduced, and mineral:matrix ratio was similar to that in wild-type mice [43]. In the analysis of cortical bone in the Ocn^−/−^ mouse line generated by Williams’ group using FTIRM, the collagen crosslink maturity and carbonate:phosphate ratio were increased relative to their control littermates, and there were no significant differences in the mineral:matrix ratio, crystallinity, or acid phosphatase level [39]. In the analysis of cortical bone in our Ocn^−/−^ mouse line using Raman microscopy, we found no significant differences in the mineral:matrix ratio, carbonate:phosphate ratio, collagen crosslink maturity, or remodeling index compared with wild-type mice [40]. Therefore, the bone quality of the three Ocn^−/−^ mouse lines analyzed by FTIRM or Raman microscopy was also inconsistent, which is unlikely to be due to the genetic background (Table 1). 

The Ocn^−/−^ mouse line in the C57BL/6J genetic background by Karsenty’s group was also examined by small-angle X-ray scattering (SAXS) analysis, which provides information on the shape of bone apatite particles, but not the crystallographic orientation (atomic arrangement). In the cortical bone of the Ocn^−/−^ mouse line, the crystal thickness was reduced, the shape was changed from rod-like to plate-like, and the crystal orientation was disorganized [44] (Table 1). 

We examined the process of mineralization by transmission electron microscopy, collagen alignment by birefringence measurements, and *c*-axis orientation of biological apatite (BAp) by a microbeam X-ray diffraction (μXRD) system, which provides information on the atomic arrangement of crystalline apatite [40]. Ocn was located in intrafibrillar and interfibrillar regions. The formation of mineralized nodules, which are globular assemblies of numerous needle-shaped mineral crystals, in Ocn^−/−^ mice was similar to that in wild-type mice, suggesting that the process of mineralization was normal. The alignment of collagen fibers was similar to that in wild-type mice, being parallel to the longitudinal direction of long bone at the diaphysis. Although the size of BAp crystallites was normal in Ocn^−/−^ mice, the crystallographic *c*-axis orientation of BAp, which is normally parallel to the orientation of collagen fibers, was highly disrupted in Ocn^−/−^ mice [40]. In several mouse models and bone types, including *oim*/*oim* osteogenesis imperfecta, c-src^−/−^ osteopetrosis, melanoma metastases, unloading, and regenerating long bones, the alignment of collagen fibers is disrupted, but BAp alignment is still parallel to the orientation of collagen fibers [45,46,47,48,49]. Thus, Ocn^−/−^ mice are the first case in which the alignment of collagen fibers was normal, but BAp alignment was disrupted [40]. Therefore, Ocn is essential for the alignment of BAp parallel to the collagen fibrils (Figure 1). 

## 5. Ocn and Bone Strength

The bone strength of Ocn^−/−^ mouse lines is also inconsistent. In the four-point bending test, the yield energy was higher in the Ocn^−/−^ mouse line in the 12Sv;C57BL/6J mixed genetic background generated by Karsenty’s group than that in wild-type mice [15]. The hardness of cortical bone was higher in the Ocn^−/−^ mouse line in the C57BL/6J genetic background generated by Karsenty’s group than that in wild-type mice in the nanoindentation test [43]. In the three-point bending test, the yield load and maximum load were also higher in the same Ocn^−/−^ mouse line than in wild-type mice [37]. In contrast, the maximum bending moment was lower in the same Ocn^−/−^ mouse line than in wild-type mice in the three-point bending test [38]. Moreover, on fatigue loading (cyclic loading) of the same Ocn^−/−^ mouse line using four-point bending, Ocn^−/−^ mice did not exhibit a significant difference from wild-type mice, whereas the double knockout mice with osteopontin (Opn) demonstrated increased stiffness, reduced energy dissipation, and higher post-fatigue creep rate [50]. As a result, Ocn^−/−^ mice, Opn^−/−^ mice, and Ocn^−/−^ Opn^−/−^ mice exhibited increased linear microcracks compared with wild-type mice, which had more diffuse damage. This suggested that the absence of these proteins attenuates the plasticity of bone and its ability to resist cyclic loading. The authors suggested that Ocn and Opn function via ionic interaction of their charged side chains with the charged surfaces of BAp [50]. Therefore, the bone strength results of the Ocn^−/−^ mice by Karsenty’s group are inconsistent even in the same genetic background (Table 1). The maximum force and stiffness were higher in Ocn^−/−^ rats than in wild-type rats in the three-point bending test [41]. 

The ultimate force, energy to ultimate force, and stiffness by the four-point bending test in the Ocn^−/−^ mouse line by Williams’ group were similar to those in wild-type mice [39]. The three-point bending test revealed no difference between our Ocn^−/−^ mouse line and wild-type mice in the maximum load, displacement, stiffness, and energy to failure [40]. However, Young’s modulus was significantly lower in our Ocn^−/−^ mouse line than in wild-type mice in nanoindentation testing along the longitudinal bone direction, and Young’s modulus was strongly and solely influenced by the orientation of BAp, but not by the orientation of collagen or BMD [40] (Table 1). These findings indicate that Ocn is required for bone strength in the longitudinal direction of long bones by adjusting the orientation of BAp parallel to collagen fibrils, and is involved in bone strength in the bone tangential direction by increasing the energy dissipation and plasticity of bone [40,50] (Figure 1). The important role of Ocn in bone strength will be the same in humans. Thus, the deficiency of vitamin K, which is required for carboxylation of the three glutamic acids of Ocn, will reduce bone strength in humans. 

## 6. Ocn and Glucose Metabolism

The Ocn^−/−^ mouse line in the 12Sv;C57BL/6J mixed genetic background generated by Karsenty’s group demonstrated impaired glucose metabolism, including increased blood glucose, impaired glucose tolerance test (GTT) and insulin tolerance test (ITT), reduced β-cell mass and insulin content in the pancreas, and increased fat mass [16]. In contrast, general knockout or osteoblast-specific knockout of *Ptprv*/*Esp*, which encodes osteotesticular protein tyrosine phosphatase (OST-PTP) and is predominantly expressed in bone, testis, and ovary, resulted in reduced blood glucose, increased serum insulin, improved the GTT and ITT, larger islets and increased β-cell proliferation in the pancreas, increased insulin sensitivity, and a decrease in visceral fat in mice [16,51,52]. Furthermore, overexpression of *Ptprv* in osteoblasts increased blood glucose, reduced serum insulin, and impaired the GTT and ITT [16]. Moreover, the glucose metabolism phenotypes of *Ptprv*^−/−^ mice were normalized by the heterozygous deletion of Ocn [16]. Carboxylated Ocn has a high affinity for Ca^2+^, whereas uncarboxylated Ocn has no affinity for Ca^2+^ and enters into the circulation [3]. Thus, uncarboxylated Ocn was considered to function as a hormone that regulates glucose metabolism, and Ptprv regulates glucose metabolism through the posttranslational modification of Ocn [16]. The following reports described how Ptprv regulates Ocn carboxylation. Ptprv inhibited insulin signaling in osteoblasts by dephosphorylating the insulin receptor. Insulin signaling inhibited FoxO1, which induced osteoprotegerin (Opg) expression to inhibit bone resorption [53,54]. Thus, Ptprv induced Opg expression through FoxO1 activation by inhibiting insulin signaling, resulting in reduced bone resorption. In the process of bone resorption, carboxylated Ocn in bone is considered to be uncarboxylated in acidic resorption lacunae, and the uncarboxylated Ocn enters into the circulation and functions as a hormone that regulates insulin secretion and insulin sensitivity [53]. Therefore, Ptprv was considered to impair glucose metabolism by reducing uncarboxylated Ocn through the inhibition of bone resorption [53]. Moreover, Atf4 was reported to impair glucose metabolism through the induction of Ptprv expression in osteoblasts, and T-cell protein tyrosine phosphatase (TC-PTP), in addition to Ptprv, was also reported to impair glucose metabolism by inhibiting insulin signaling in osteoblasts by dephosphorylating the insulin receptor, which leads to the reduction in bone resorption and uncarboxylated osteocalcin [55,56]. All of these reports were based on the Ocn^−/−^ mouse line in the 129Sv;C57BL/6J mixed genetic background generated by Karsenty’s group that exhibited impaired glucose metabolism. 

In contrast to these reports, in the Ocn^−/−^ mouse line by Williams’ group, the levels of random fed blood glucose and fasted blood glucose were similar to those in wild-type mice in both males and females [39]. The levels of random fed blood glucose and HbA1c at 11 weeks to 18 months of age in our Ocn^−/−^ mouse line were also similar to those in wild-type mice in both males and females. Moreover, subcutaneous and visceral fat masses were similar between our Ocn^−/−^ mouse line and wild-type mice in both males and females. The GTT and ITT using mice fed a normal diet or high-fat diet were normal for our Ocn^−/−^ mouse line at 14 weeks to 18 months of age for both males and females [40]. The fasting blood glucose, fat mass, and GTT in Ocn^−/−^ rats were similar to those in wild-type rats, although the ITT significantly improved in Ocn^−/−^ rats compared with wild-type rats [41]. These suggest that Ocn is not a hormone that physiologically regulates glucose metabolism [57].

Why the Ocn^−/−^ mouse line in the 129Sv;C57BL/6J mixed genetic background generated by Karsenty’s group exhibited impaired glucose metabolism needs to be investigated [58,59]. As there are inherent differences in glucose metabolism between 129 and C57BL/6 mouse lines [60,61,62,63,64,65], it is important to analyze the mice in a pure genetic background to reduce the variation in the data. C57BL/6J mice exhibit impaired insulin secretion and glucose intolerance, the C57BL/6J strain, but not the 129Sv or C57BL/6N strain, has a mutation in the nicotinamide nucleotide transhydrogenase (*Nnt*) gene, and the impaired insulin secretion and glucose intolerance are rescued by the transgenic expression of *Nnt* [66,67,68,69,70] (Mouse Genome Project: Sanger institute, https://www.sanger.ac.uk/data/mouse-genomes-project/). Thus, *Nnt*, which influences β cell insulin secretion in the pancreas [67,70], is one of the responsible proteins for the impaired insulin secretion and glucose intolerance in the C57BL/6J strain. If Ocn^+/−^ littermates in the 129Sv;C57BL/6J mixed genetic background were continuously mated to obtain Ocn^+/+^ and Ocn^−/−^ mice, their genotypes are *Nnt*^+/+^, *Nnt*^+/−^, or *Nnt*^−/−^. Therefore, control wild-type mice with the *Nnt*^−/−^ genotype should exhibit impaired insulin secretion and glucose intolerance, and the glucose metabolism in Ocn^−/−^ mice with the *Nnt*^−/−^ genotype should be more markedly impaired than that in Ocn^−/−^ mice with the *Nnt*^+/+^ genotype. However, in the analyses of the Ocn^−/−^ mouse line in the 129Sv;C57BL/6J mixed genetic background, the standard deviation in the values of blood glucose, GTT, and ITT in both the control wild-type and Ocn^−/−^ mice was small [16]. Thus, the *Nnt* genotype was the same in the wild-type mouse group and Ocn^−/−^ mouse group. If mating was performed correctly, this should not occur. As such, *Nnt*^+/+^Ocn^+/+^ littermates may have been mated to expand control wild-type mice, and *Nnt*^−/−^Ocn^−/−^ littermates may have been mated to expand Ocn^−/−^ mice. If it is was case, the impaired glucose metabolism in Ocn^−/−^ mice was likely caused by the *Nnt* mutation, not by the absence of Ocn. C57BL/6J mice, but not C57BL/6N mice, have impaired glucose metabolism, and there are many genetic variants between C57BL/6J and C57BL/6N substrains in addition to *Nnt* mutation [68,71]. Some of the variant genes may also be involved in the impaired glucose metabolism in C57BL/6J mice because C57BL/6J and C57BL/6N mice have a similar β-cell mass in the pancreas [72], but it was reduced in the Ocn^−/−^ mice generated by Karsenty’s group [16]. 

The abnormal (improved) glucose metabolism in *Ptprv*_osb_^−/−^ mice may also be explained by the genetic background. *Ptprv*_osb_^−/−^ mice were analyzed in a 129Sv;FVB mixed genetic background [16,55,73], which does not have the *Nnt* mutation [70] (Mouse Genome Project: Sanger Institute, https://www.sanger.ac.uk/data/mouse-genomes-project/). Although 129Sv and FVB strains do not have *Nnt* mutation, the levels of *Nnt* expression and insulin secretion are much higher in the FVB strain than those in the 129Sv strain [70]. Thus, if the crossing was inappropriately performed to expand wild-type and *Ptprv*_osb_^−/−^ mice, *Ptprv*_osb_^−/−^ mice with high *Nnt* expression and insulin secretion, and wild-type mice with low *Nnt* expression and insulin secretion may be generated, and it may explain abnormal (improved) glucose metabolism in *Ptprv*_osb_^−/−^ mice. Further, the abnormal (improved) glucose metabolism in *Ptprv*_osb_^−/−^ mice was reversed by lacking one allele of Ocn [16]. As Ocn^+/−^ or Ocn^−/−^ mice were maintained in the 129Sv;C57BL/6J mixed genetic background [15,16], in which *Nnt* expression and insulin secretion are extremely low [70], glucose metabolism in *Ptprv*_osb_^−/−^Ocn^+/−^ mice in a 129Sv;FVB;C57BL/6J mixed genetic background will be impaired as compared with that in *Ptprv*_osb_^−/−^ mice, explaining the reversal of glucose metabolism in *Ptprv*_osb_^−/−^Ocn^+/−^ mice. Thus, it is extremely important to consider the phenotypic differences among strains in the analyses of gene-targeted mice and compare the gene-targeted mice in the same genetic background. 

## 7. Association of Ocn with Exercise and Glucose Metabolism 

Many clinical studies demonstrating the positive or negative association of serum total Ocn or uncarboxylated Ocn with glucose metabolism or cardiovascular risk have been reported [74,75,76,77,78,79,80,81,82,83,84,85,86,87,88,89,90,91,92,93]. When bone formation is increased, osteoblasts increase Ocn production, and serum Ocn increases. In this situation, total Ocn, carboxylated Ocn, and uncarboxylated Ocn levels increase in the serum. When bone resorption is increased, Ocn in bone is uncarboxylated in acidic resorption lacunae, and uncarboxylated Ocn enters into the circulation. Thus, uncarboxylated Ocn in the serum increases during both bone formation and resorption. Exercise increases bone formation, and the total Ocn, carboxylated Ocn, and uncarboxylated Ocn levels. It also improves glucose metabolism, which reduces cardiovascular risk. Therefore, the positive association of serum total Ocn or uncarboxylated Ocn with improved glucose metabolism or reduced cardiovascular risk can be explained by the exercise-induced Ocn [4,40]. 

## 8. Function of Ocn in Testosterone Synthesis and Muscle Mass

When Ocn^−/−^ male mice in the 129Sv genetic background generated by Karsenty’s group were mated with wild-type female mice, the number of pups per litter was smaller than when wild-type males were mated with wild-type females, and the frequencies of litters also decreased. The Ocn^−/−^ male mice had smaller testes and fewer sperm than wild-type male mice, germ cell apoptosis was increased, the levels of serum testosterone were markedly reduced, and the expression of genes required for testosterone biosynthesis was markedly reduced [17]. In contrast, when Ocn^−/−^ males generated by Williams’ group were crossed with C57BL/6J females, the number of pups per litter and the frequencies of litters were similar to those in the crosses of wild-type control littermates with C57BL/6J females. Furthermore, the level of serum testosterone was similar to that in wild-type control littermates [39]. Our Ocn^−/−^ males also had a similar testis weight to wild-type mice. The number of sperm, the frequencies of sperm with acrosomal defects, the levels of serum testosterone, and the expression of the genes related to testosterone biosynthesis were similar to those in wild-type mice. The testes were histologically normal, and the frequencies of germ cell apoptosis were similar to those in wild-type mice [40]. Thus, the phenotypes in the testes of the Ocn^−/−^ mice generated by Karsenty’s group were also different from those of the Ocn^−/−^ mice generated by Williams’ and our groups. 

*Gprc6a* was reported to be a receptor for uncarboxylated Ocn in Leydig cells in the testis and β cells in the pancreas [17,94]. However, it is controversial whether *Gprc6a* is involved in insulin secretion and whether it functions as a receptor for osteocalcin in Leydig cells and β cells [17,94,95,96,97,98,99]. Moreover, the phenotypes of *Gprc6a*^−/−^ mice are controversial. There are three different global *Gprc6a*^−/−^ mouse models, in which exon 2, exon 6, or the full locus is deleted. Mice with the deletion of exon 2 exhibited impaired glucose metabolism and reduced testosterone and BMD, mimicking the phenotypes of the Ocn^−/−^ mice by Karsenty’s group, excluding BMD [100]. In addition, pancreas- or Leydig cell-specific deletion of *Gprc6a* resulted in similar phenotypes to those observed for Ocn^−/−^ mice generated by Karsenty’s group [17,97]. In contrast, mice with the global deletion of exon 6 or the full locus of *Gprc6a* demonstrated normal glucose metabolism, fertility, and BMD [95,101,102]. Interestingly, *Gprc6a*^−/−^ mice and pancreas-specific *Gprc6a*^−/−^ mice using *Ins2*-Cre in Quarles’ group, which showed impaired glucose metabolism similar to that in Ocn^−/−^ mice, were analyzed in 129Sv;C57BL/6J mixed genetic background [97,100]. In contrast, *Gprc6a*^−/−^ mice in the Bräuner–Osborne group, which showed normal glucose metabolism, were analyzed in a C57BL/6N genetic background [102]. Thus, the discrepancy in the glucose metabolism in *Gprc6a*^−/−^ mice may also be explained by the phenotypic differences between C57BL/6J and C57BL/6N strains, because C57BL/6J but not C57BL/6N strain has *Nnt* mutation [66,67,68,69]. 

The Ocn^−/−^ mice by Karsenty’s group also had reduced muscle mass, and the average area of the muscle fibers was smaller than that in wild-type mice, and uncarboxylated Ocn promoted protein synthesis in myofibers [103]. Moreover, exercise was reported to increase circulating interleukin 6 (IL-6), which originates from muscle. Further, IL-6 was shown to induce osteoclast differentiation and bone resorption, increase the circulation of uncarboxylated Ocn, and promote the uptake and catabolism of glucose and fatty acids in myofibers during exercise in an Ocn-dependent manner [104]. 

In our Ocn^−/−^ mice, however, the weights of muscles and the average area of muscle fibers were similar to those in wild-type mice [40]. Thus, the muscle phenotypes of Ocn^−/−^ mice generated by Karsenty’s group were also not reproduced. 

## 9. Conclusions

Ocn plays an essential role in bone by regulating the alignment of BAp parallel to collagen fibrils and is required for bone strength in the longitudinal direction of the long bone [40]. However, the inhibition of bone formation by Ocn, which was demonstrated by Ocn^−/−^ mice generated by Karsenty’s group [15], was not reproduced [39,40]. Ocn was previously reported to function as a hormone that regulates insulin secretion in the pancreas, testosterone synthesis in the testis, and muscle mass based on the analysis of the Ocn^−/−^ mice by Karsenty’s group [16,17,103]. However, these hormonal functions were not reproduced in two independently generated Ocn^−/−^ mouse lines or Ocn^−/−^ rats [39,40,41] (Figure 2). The phenotypic differences among the strains are one of the explanations of the controversy. Ocn was also reported to play a role in brain development and to regulate anxiety and cognition in adult mice based on the analysis of Ocn^−/−^ mice by Karsenty’s group [19,105]. These findings also need to be investigated by other groups to confirm their reproducibility. Furthermore, the administration of a large amount of Ocn improved glucose metabolism, prevented anxiety and depression, and improved memory [19,105,106]. These effects of exogenous Ocn also need to be investigated by many groups to confirm their reproducibility. 

## Figures and Tables

**Figure 1 ijms-21-07513-f001:**
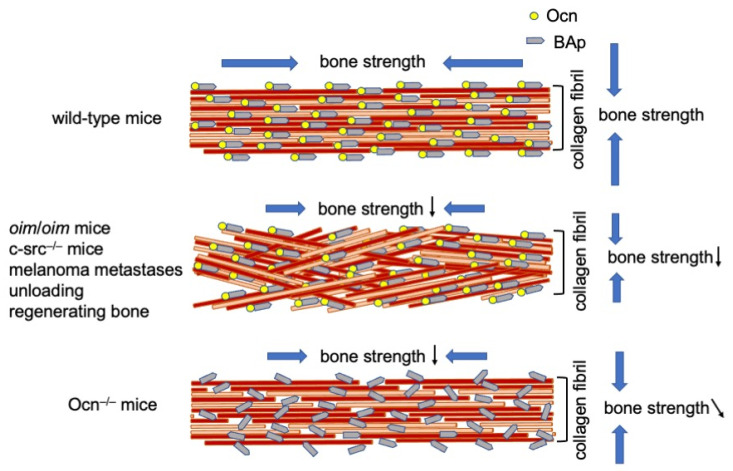
Alignment of collagen fibers and BAp. In wild-type mice, the orientation of collagen fibers is parallel to the longitudinal direction of long bone at the diaphysis, and that of BAp is parallel to the collagen fibers. In the mouse models and bone types, including *oim*/*oim* osteogenesis imperfecta, c-src^−/−^ osteopetrosis, melanoma metastases, unloading, and regenerating long bones, the alignment of collagen fibers is disrupted, but that of BAp is still parallel to the orientation of collagen fibers. The bone strength is reduced in both longitudinal and vertical directions of the long bone. In Ocn^−/−^ mice, the orientation of collagen fibers is parallel to the longitudinal direction of the long bone, but the alignment of BAp is disrupted. The bone strength is reduced in the longitudinal direction of the long bone, and that in the tangential direction is slightly reduced. ↓: reduced, ↘: slightly reduced.

**Figure 2 ijms-21-07513-f002:**
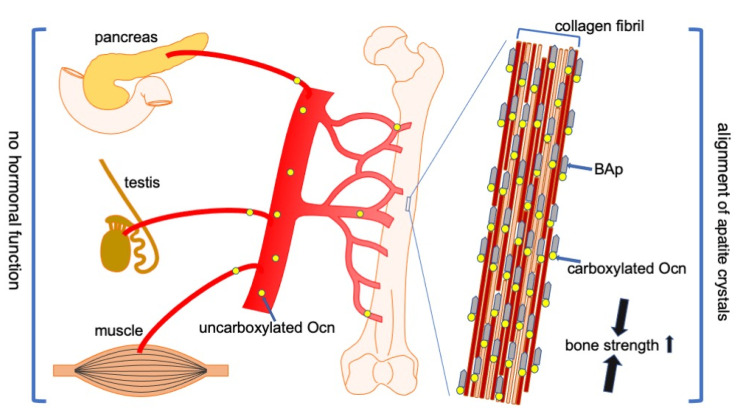
Functions of Ocn in bone, pancreas, testis, and muscle. Carboxylated Ocn is required for the alignment of BAp parallel to the collagen fibers and optimal bone strength. However, two newly generated Ocn^−/−^ mouse lines and Ocn^−/−^ rats did not exhibit the impaired glucose metabolism, reduced testosterone synthesis and spermatogenesis, and reduced muscle mass observed in the Ocn^−/−^ mouse line generated by Karsenty’s group. Thus, uncarboxylated Ocn does not physiologically function as a hormone that regulates glucose metabolism in the pancreas, testosterone synthesis in testis, or muscle mass. ↑: Bone strength is increased by carboxylated Ocn.

**Table 1 ijms-21-07513-t001:** Comparison of bone quantity and quality in osteocalcin (Ocn)^−/−^ mouse lines.

	Karsenty’s Group		Williams’ Group	Our Group
Method	Deletion of *Bglap* and *Bglap2* Using ES Cells		Deletion of *Bglap* and *Bglap2* by CRISPR/Cas9	Deletion of *Bglap* and *Bglap2* Using ES Cells
genetic background	129Sv;C57BL/6J	C57BL/6J	C57BL/6J;C3H	C57BL/6N
cortical bone	↑↑	→	→	→
trabecular bone	↑↑	↑	→	→
bone formation	↑	↑	nd	→
bone resorption	↑	→	nd	→
ovariectomy	bone resorption Ocn^−/−^ > wild-type	nd	nd	similar to wild-type
cortical BMD	nd	→ or ↓	→	→
trabecular BMD	nd	→	→	→
crystallinity	↓	↑	→	→
mineral:matrix ratio	→	→	→	→
carbonate:phos-phate ratio	→	↓	↑	→
collagen maturity	nd	nd	↑	→
size or shape of BAp	nd	thin, plate-like	nd	Size →
bone strength	↑	↑ or ↓	→	↓

nd: not done. ↑↑: markedly increased, ↑: increased, →: no change, and ↓: reduced as compared with control wild-type mice.

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
