# Peer review of "Functions of Osteocalcin in Bone, Pancreas, Testis, and Muscle"

_ijms, 2020, doi:10.3390/ijms21207513_

Round 1

Reviewer 1 Report

This manuscript reviews current knowledge on the potential physiological roles of osteocalcin (Ocn) on bone, testes, pancreas and muscle. While the subject is interesting and controversies in the field justify a review, some insight and conceptualization into the broad implications and relevance of identifying or confirming the physiological roles of Ocn would be helpful and allow the reader to obtain a wider vision of the subject. As it is, the review seems mostly a rebuttal of the Karsenty's group results. The review would be more informative if the problems and questions identified were approached in a more comprehensive way, namely by acknowledging the limitations of existing models and identifying other models that may contribute to elucidate current and other questions. Moreover, the author should also consider otehr potential explanations for the discrepancies observed in different studies. For instance, is the expression of Ocn in the KO mice completely abrogated or is there some residual expression? Can different magnitudes of Ocn expression explain or, at least, contribute to the differences observed between the 3 models? Also, how does the observation that the heterozygous deletion of Ocn in Ptprv–/–mice normalizes glucose homeostasis fits into the hypothesis of selection of different Nnt genotypes? The not random selection of the Nnt genotype doesn't apply to the Ptprv–/– mouse line which was developed in the same background? The author should elaborate more on this hypothesis. 

In fig. 1, why isn't BAp represented similarly in the 3 types of condition? Does the picture represent the same molecular species in the 3 cases or not?

The link to the genome project has a spelling mistake ("geome") that prevents accession.

Author Response

Response to the reviewer 1

Reviewer 1
This manuscript reviews current knowledge on the potential physiological roles of osteocalcin (Ocn) on bone, testes, pancreas and muscle. While the subject is interesting and controversies in the field justify a review, some insight and conceptualization into the broad implications and relevance of identifying or confirming the physiological roles of Ocn would be helpful and allow the reader to obtain a wider vision of the subject.

I rewrote the abstract by deleting detailed description of the phenotypes of Ocn–/– mice and adding more understandable description of the physiological roles of Ocn. Now the readers will be able to obtain a wider and clear vision of this subject.

As it is, the review seems mostly a rebuttal of the Karsenty's group results. The review would be more informative if the problems and questions identified were approached in a more comprehensive way, namely by acknowledging the limitations of existing models and identifying other models that may contribute to elucidate current and other questions. Moreover, the author should also consider other potential explanations for the discrepancies observed in different studies. For instance, is the expression of Ocn in the KO mice completely abrogated or is there some residual expression? Can different magnitudes of Ocn expression explain or, at least, contribute to the differences observed between the 3 models?

Ocn protein was completely absent in the three Ocn–/– mouse lines and Ocn–/– rats. I described it on page 3 (lines 116-117). I suggested that the generation of Ocn–/– mouse line in a pure 129Sv genetic background may help to resolve the controversy on page 3 (lines 121-122). I also suggested that the impaired or improved glucose metabolism may be caused by the phenotypic differences among strains, and emphasized the importance to consider the phenotypic differences among strains in the analyses of gene targeted mice on page 7 (lines 275-290) and page 8 (lines 330-336).

Also, how does the observation that the heterozygous deletion of Ocn in Ptprv–/–mice normalizes glucose homeostasis fits into the hypothesis of selection of different Nnt genotypes? The not random selection of the Nnt genotype doesn't apply to the Ptprv–/– mouse line which was developed in the same background? The author should elaborate more on this hypothesis. 

The abnormal (improved) glucose metabolism in Ptprvosb/– mice may also be explained by the genetic background. Ptprvosb/– mice were analyzed in a 129Sv;FVB mixed genetic background(Cell 130:456-469, 2007; J Clin Invest 119:2807-2817, 2009; Dev Dyn 224:245-251, 2002), which does not have the Nnt mutation (Mouse Genome Project : Sanger institute, https://www.sanger.ac.uk/data/mouse-genomes-project/) (Diabetologia 50: 2475-2485, 2007). Although 129Sv and FVB strains do not have Nnt mutation, the levels of Nnt expression and insulin secretion are much higher in FVB strain than those in 129Sv strain (Diabetologia 50: 2475-2485, 2007). Thus, the crossing was inappropriately performed to expand wild-type and Ptprvosb/– mice, Ptprvosb/– mice with high Nnt expression and insulin secretion and wild-type mice with low Nnt expression and insulin secretion may be generated, and it may explain abnormal (improved) glucose metabolism in Ptprvosb/– mice. Further, the abnormal (improved) glucose metabolism in Ptprvosb/– mice was reversed by lacking one allele of Ocn(Cell 130:456-469, 2007). As Ocn+/– or Ocn–/– mice were maintained in the 129Sv;C57BL/6J mixed genetic background, in which Nnt expression and insulin secretion are extremely low (Diabetologia 50: 2475-2485, 2007), glucose metabolism in Ptprvosb/–Ocn+/– mice in a 129Sv;FVB;C57BL/6J mixed genetic background will be impaired as compared with that in Ptprvosb/– mice, explaining the reversal of glucose metabolism in Ptprvosb/–Ocn+/– mice. Thus, it is extremely important to consider the phenotypic differences among strains in the analyses of gene targeted mice and to compare the gene targeted mice in the same genetic background.

I described it in the section 5 (page 7, lines 275-290).            

Interestingly, Gprc6a–/– mice and pancreas-specific Gprc6a–/– mice using Ins2-Cre in Quarles’ group, which showed impaired glucose metabolism similar to that in Ocn–/– mice,  were analyzed in 129Sv;C57BL/6J mixed genetic background (PLOS one 3: e3858, 2008; Endocrinology 157: 1866-1880, 2016). In contrast, Gprc6a–/– mice in Bräuner-Osborne group, which showed normal glucose metabolism, were analyzed in a C57BL/6N genetic background (Sci Rep 9: 5995, 2019). Thus, the discrepancy in the glucose metabolism in Gprc6a–/– mice may also be explained by the phenotypic differences between C57BL/6J and C57BL/6N strains, because C57BL/6J but not C57BL/6N strain has Nntmutation (Diabetologia 48: 675-686, 2005; Diabetes 55: 2153-2156, 2006; Diabetes 65: 25-33, 2016; Obesity 18: 1902-1905, 2010).

I described it in section 7 (page 8, lines 330-336).

I also described that the phenotypic differences among the strains are one of the explanations of the controversy in the conclusion (page 9, lines 361-362).

In fig. 1, why isn't BAp represented similarly in the 3 types of condition? Does the picture represent the same molecular species in the 3 cases or not?

I showed Ocn, BAp, and collagen fibril in the 3 cases in Fig. 1.

The link to the genome project has a spelling mistake ("geome") that prevents accession.

Thank you for pointing out the misspelling.

Reviewer 2 Report

This study reported that Ocn plays an important role in bone, but its hormonal functions in the pancreas, testis, and muscle were not reproduced. A revision is suggested.

  1. Please discuss the clinical implications in this study.
  2. Fig2 needs a revision. Blood vessels do not attach to bones.
  3. The physiological roles of Ocn in human need to be further addressed.

Author Response

Response to the reviewer 2

Reviewer 2

This study reported that Ocn plays an important role in bone, but its hormonal functions in the pancreas, testis, and muscle were not reproduced. A revision is suggested.

  1. Please discuss the clinical implications in this study.

I added the following sentences in section 5 to discuss the clinical implication of Ocn in bone (Page 6, lines 204-207):

The important role of Ocn in the bone strength will be the same in humans. Thus, the deficiency of vitamin K, which is required for carboxylation of the three glutamic acids of Ocn, will reduce the bone strength in humans.

Further, I added the following sentence in the last part of abstract to mention the connection of serum Ocn and glucose metabolism:

Clinically, serum Ocn is a marker for bone formation, and exercise increases bone formation and improves glucose metabolism, making a connection of Ocn and glucose metabolism.

  1. Fig2 needs a revision. Blood vessels do not attach to bones.

Thank you for indicating it. I modified Fig. 2 and the graphic abstract.

  1. The physiological roles of Ocn in human need to be further addressed.

The physiological role of Ocn in human will be the same with mice, which is the maintenance of the bone strength by adjusting the orientation of BAp parallel to collagen fibrils, and by increasing the energy dissipation and plasticity of bone. As described in the comment 1, I added the following sentences in the section of “Ocn and bone strength” to discuss the physiological role of Ocn (Page 6, lines 204-207):

The important role of Ocn in the bone strength will be the same in humans. Thus, the deficiency of vitamin K, which is required for carboxylation of the three glutamic acids of Ocn, will reduce the bone strength in humans.

Round 2

Reviewer 2 Report

All my questions had been well addressed. This submission is acceptable.